# Problems of Access to Services at Railway Stations in Freight Transport in the Slovak Republic

**Eva Brumercikova**  **and Adrian Sperka** *

Faculty of Operation and Economics of Transport and Communications, University of Zilina,
010 26 Zilina, Slovakia; eva.brumercikova@fpedas.uniza.sk
* Correspondence: adrian.sperka@fpedas.uniza.sk; Tel.: +421-41-513-3434

**Abstract:** The article deals with the issue of access of freight carriers to services in railway stations. With the liberalization of the rail freight market, the number of carriers is gradually increasing. In this situation, infrastructure capacity is often insufficient. Therefore, it is necessary to set the order of access to services in railway stations. The article will use the process of analytical hierarchy as one of many methods of multicriteria analysis. Four important indicators will be selected for carriers: railway stations where the service of stabling tracks can be used, railway stations where rail scales are located, railway stations where it is possible to use the services of a shunting movement from the infrastructure manager, railway stations where the carrier can be using the services of a mobile workshop. At the end of the article, the order of access to these services will be compiled according to the order of importance for railway undertakings. A significant factor will also be an approximate quantification of the performance of individual carriers passing through the selected station.

**Keywords:** multicriteria analysis; freight railway transport; infrastructure manager; railway stations

## 1. Introduction

Liberalization of railway transport is a gradual process which began at the end of the last century [1]. The one railway company operated passenger, freight transport and also railway infrastructure before liberalization. The liberalization process meant the activities of this entity had to be separated at least in accounting. The main aim of this process was to gain an overview of financing the railway transport undertakings, to make the railway transport market functioning more effective, and to open it for other railway companies. Some EU member states did not respect the liberalization regulations for a long time and they started the process late despite sanctions resulting from their decision. An example is an article by the author team of Associate Professor Nedeliakova, which discusses the synergistic effects of the liberalization of the rail transport market [2].

Each bigger intervention into the transport sector has its downsides, too. The course of the process itself has been accompanied by several problems. Among others there has been an increase in railway transport. It has been increasing directly proportionally with new carriers which entered the railway network upon fulfilling the criteria. The capacity of railway infrastructure was not built for such an increase in many cases. So, in spite of guarantees of non-discriminatory access to the railway network, in practice it often happens that the services of the infrastructure manager are accessed by a carrier whose 100% share is owned by the state in preference. This situation results in a deterioration in the quality of the services provided, both for railway undertakings and for customers. That implies it is necessary to determine the order for the access to services. It is a challenging and not always realizable process with regard to the stochastic character of the railway operation [3]. There exist some decision-making tools of mathematical statistics, namely Saaty's method (also known as Analytic Hierarchy Process—AHP) and interpretive structural modeling (ISM) analysis [4] which facilitate

decision making. It is thus a decision-making process where in a given relevant environment (in this case, the environment of railway transport) multiple subjects of decision making select a certain solution out of a bigger number of possible solutions. The decision-making process is realized through the application of a certain arranged evaluation system on the basis of certain rules. The output of this evaluation is often the optimization of various processes [5].

## 2. The Process of Railway Transport in Slovakia

The transformation process can be divided into several stages. Table 1 presents fundamental milestones of transformation of railways and their legislation.

**Table 1.** Important milestones of railway transformation in Slovakia [5,6].

| Phase | Date | Event |
|-------|------|-------|
| No. 1 | 1 January 1993 | Establishment of the Railway of the Slovak Republic (ŽSR) and Czech Railways (ČD)—based on the demise of Czechoslovakia and the Czechoslovak State Railways |
| | 30 September 1993 | The act on ŽSR was adopted (258/1993 Z. z.) |
| | 1 January 1997 | Divisional arrangement of the ŽSR—related to the transformation of the company into a trading company |
| No. 2 | 14 June 2001 | Parliament approved the law on Železničná Spoločnosť Slovensko, a.s. (ZSSK). |
| No. 3 | 1 January 2002 | The railway sector was divided into ŽSR, which became the infrastructure manager, and ZSSK, a.s., which became the national passenger and freight carrier |
| | 1 January 2005 | The railway sector was divided into ŽSR, which became the infrastructure manager, ZSSK, a.s., which became the national passenger carrier, and Železničná spoločnosť Cargo Slovakia, a.s. (ZSSK CARGO) |
| | 1 June 2014 | Transformation ZSSK CARGO—spin-offs subsidiaries |

The liberalization process of freight transport has been influenced by the historical milestone of January 1999, when the company VSŽ Oceľ, s.r.o., Košice as the first entity asked for a facility to utilize a railway communication. Based on an agreement between the undertaking and ŽSR, the operation started on 1 February 2000 [4]. The object in question was the transport of limestone from Turňa nad Bodvou to Haniska near Košice [6]. The railway communication was actually made accessible on 1 January 2003. Since then, under the applicable legislation, ŽSR has been obliged to allow other entities, not only ZSSK, a. s., to use the railway infrastructure and its service facilities. In that time there existed several smaller railway undertakings in railway freight transport, however, they focused on carriage of some kinds of goods only. Other potential railway undertakings in Slovakia were discouraged mainly by a high price for the utilization of the railway communication. In that time the state guaranteed a subsidy covering just a fraction of costs associated with the utilization of the railway communication [4].

All changes introduced in Table 1 were made in the context of a legal framework for a unified arrangement of relationships among transport undertakings, the infrastructure manager and the state under the conditions of a Single European Rail Area which is controlled with so-called railway directives [6]. They are issued in four stages of reform and are presented in Table 2.

**Table 2.** Reform of the railway sector in the EU [6].

| Railway Package | Objective of the Reform |
|---|---|
| First railway package | The first directive imposes the separation of railways from the state |
| | The second directive imposes a division of rail transport activities into infrastructure-related activities and activities involving the operation of rail transport |
| | The third directive imposes infrastructure regulation and the licensing of railway undertakings |
| Second railway package | The aim is to create a legally and technically integrated European railway |
| Third railway package | Introduces measures to open international passenger transport services in the framework of EU free competition |
| Fourth railway package | The aim is to increase the quality and efficiency of railway transport services by removing remaining barriers to the market |

The implementation of these conditions in the national legislation was one of the conditions for Slovakia to enter the EU member states.

## 3. Evaluation of the Availability of Railway Stations Using the AHP Method

The decision-making process is a process of choice from multiple possible solutions [7]. In the strict sense of the world it may be characterized as a search and a selection of suitable variants of a solution of an arisen problem [8].

It is a process of solving decision-making problems, and it comprises two parts [8]:

- the evaluation,
- the selection of an optimal variant.

Figure 1 shows a general decision-making process using a cyclically repeated chart.

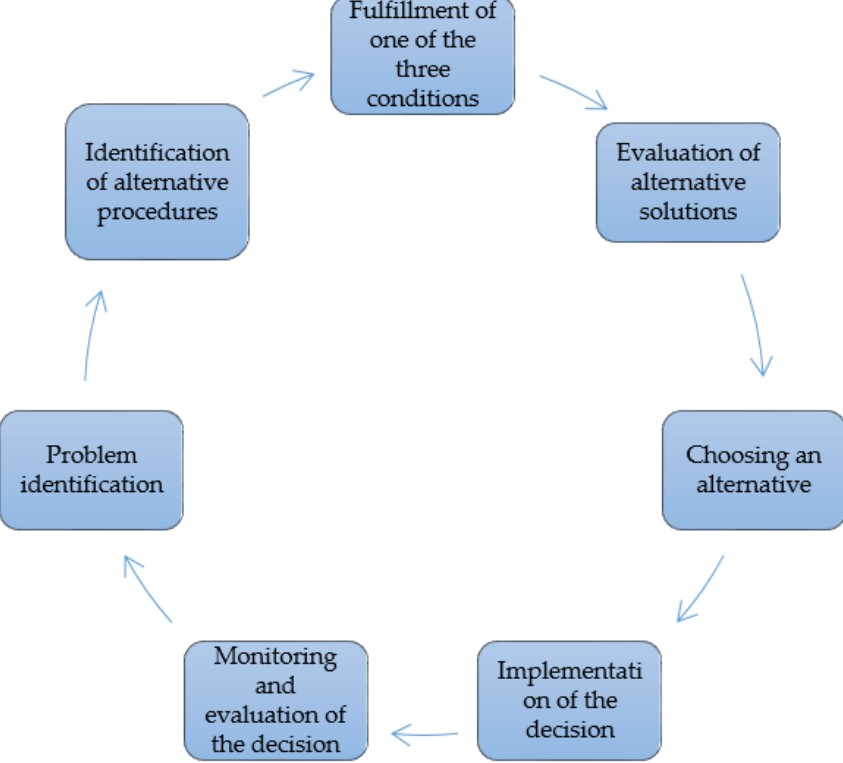

**Figure 1.** Decision-making process [9].

We can see on the chart that the process of decision making starts with the identification of a problem (in this case it is the access to railway station services which are offered by the infrastructure manager). In the next step it is the determination of alternative procedures and meeting one of three requirements (it is the requirement for which the intensity is expressed using the rate of utilizing a service by a carrier). In the closing part of the theoretical stage, the alternative solutions (in our case, the urgency to realize modernization or reconstruction measures focused on the most utilized service) are evaluated. Then one of the alternatives is selected; afterwards the decisions are implemented and at the end these decisions are reviewed and evaluated.

### 3.1. Evaluated Indicators

Each railway station in the ŽSR network has its role and a considerable mission. Some railway stations fulfil bigger tasks and they are equipped with proper resources for that; others are in charge of smaller tasks with a smaller scope of the operation's work [10].

The conditions of utilizing the railway network are defined by railway stations according to selected criteria [11,12]:

- operating control points (railway stations) with a closure of a transport service,
- railway stations where streamlined technological procedures of acts can be applied in case of freight transport trains,
- transport points where a service of stabling sidings can be utilized,
- transport points where shunting services provided by the infrastructure manager can be utilized,
- railway stations with rail weighbridges,
- railway stations with a siding,
- charging for the access to railway infrastructure and service facilities.

Besides the criteria mentioned above there are also other criteria important from the point of view of carriers, such as usage of mobile workshops, a technology of delivery and acceptance of serried trains at border transit stations, or dispatch privileges of railway stations [13].

As part of the evaluation using the AHP method, four criteria will be selected:

- transport points where a service of stabling sidings can be utilized,
- railway stations where rail weighbridges are placed,
- transport points where shunting services provided by the infrastructure manager can be utilized,
- the possibility to utilize services of a mobile workshop.

### 3.2. Evaluation Bodies

Evaluative entities will be represented by railway undertakings which actively utilize the services of railway stations being evaluated. There will be two railway undertakings with an open structure of financing (Prvá Slovenská Železničná, a.s., and Metrans Danubia, a.s.), and one railway undertaking with its shares owned by the state only (Železničná spoločnosť Cargo Slovakia, a.s.). Their brief characteristics are presented in Table 3.

**Table 3.** Characteristics of railway undertakings.

| The Name of Railway Undertaking | Brief Characteristics |
| --- | --- |
| ZSSK CARGO | Specializes in the transport of block trains, logistics train and for the transport of individual wagon consignments |
| | Transports all kinds of goods, the transport of iron ore predominates |
| | is a network carrier |

**Table 3.** *Cont.*

| The Name of Railway Undertaking | Brief Characteristics |
|---|---|
| Prvá Slovenská Železničná | Operates only block trains |
| | Obtained licenses to operate railway transport in the Czech Republic, Hungary and Romania |
| | The portfolio of transported commodities is wide, but priorities include wood, malt, maize and slag |
| Metrans Danubia | Specializes mainly in the transport of containers and empty platform wagons Has subsidiaries throughout Central and Western Europe in Slovakia the company owns an intermodal transport terminal in Dunajská Streda and due to these their transport routes are realized mainly on the line Komárno—Bratislava |

Each of the carriers mentioned above utilizes services offered by the infrastructure manager to a different extent. The factor of their importance for each carrier is described in an illustrative example in the following section.

### 3.3. Application of AHP Method in Railway Transport

There exist multiple different methods which basically feature the same principle—the assessment of several variants of solution of a given problem according to selected criteria and a set order of individual variants. Particular methods differ in how the weight of criteria are determined and the degree to which the individual variants of solution meet the selected criteria is evaluated numerically [14].

In the case of the AHP method, the comparison of criteria as well as of individual variants is based on a so-called expert estimate in which experts on a given field of study compare mutual impacts of two factors [15]. These evaluations are presented in Table 4.

**Table 4.** Determination of weights [16].

| Scale | Description |
|---|---|
| 1 | Elements are just as important |
| 2 | A row element is very slightly more significant than the column element |
| 3 | A row element is slightly more significant than the column element |
| 4 | A row element is more important than the column element |
| 5 | A row element is much more important than the column element |
| 6 | A row element is demonstratively more significant than the column element |
| 7 | A row element is demonstratively much more significant than the column element |
| 8 | A row element is much more important than the column element |
| 9 | A row element is indisputably more significant than a column element |

For a better arrangement, see Table 5 where the evaluated indicators and evaluative elements are marked and sorted with letters.

**Table 5.** Identification of criteria and their users.

| Letter | Rated Services | Letter | Railway Undertakings Using Rated Services |
|---|---|---|---|
| A | Railway stations where the service of stabling tracks can be use | A | ZSSK Cargo |
| B | Railway stations where rail scales are located | B | PSŽ |

**Table 5.** *Cont.*

| Letter | Rated Services | Letter | Railway Undertakings Using Rated Services |
|--------|----------------|--------|-------------------------------------------|
| C | Railway stations where it is possible to use the services of a shunting movement from the infrastructure manager | C | Metrans Danubia |
| D | Railway stations where the carrier can use the services of a mobile workshop | | |

In the first stage of the first step, individual services (criteria) are compared first. They can be compared on different bases. The best way, however, is to compare them on the basis of frequency of their utilization. For different railway stations in the network, however, there exists a different frequency. Thus, it is appropriate to choose one station where the importance of its services is determined by the frequency of their utilization. It will be Štúrovo station in our case, which is an important border transit station between Slovakia and Hungary. Table 6 presents a pair-wise comparison of services criteria at Štúrovo railway station.

**Table 6.** Pair-wise comparison of criteria.

| Criterion | Weight | Criterion |
|-----------|--------|-----------|
| A | 9 8 7 6 5 4 3 2 1 2 3 4 5 6 7 8 9 | B |
| A | 9 8 7 6 5 4 3 2 1 2 3 4 5 6 7 8 9 | C |
| A | 9 8 7 6 5 4 3 2 1 2 3 4 5 6 7 8 9 | D |
| B | 9 8 7 6 5 4 3 2 1 2 3 4 5 6 7 8 9 | C |
| B | 9 8 7 6 5 4 3 2 1 2 3 4 5 6 7 8 9 | D |
| C | 9 8 7 6 5 4 3 2 1 2 3 4 5 6 7 8 9 | D |

The basis of the AHP method is the recording of individually selected significance values which were compared among alternatives in evaluative forms (Table 6) in a so-called Saaty's decision matrix [17,18]. The number of individual comparisons can then be calculated using Equation (1).

$$number\ of\ comparisons = \frac{n \times (n-1)}{2} \tag{1}$$

In our case *n* represents the number of elements we want to compare. In the result there are six comparisons in total, which is proved in Table 6. In Table 6, red is used for pair-wise comparison of the criteria listed in Table 5. In Table 7, red is used to express the importance for pair-wise comparison of criteria by variant. In both cases, the red color emphasizes the degree of importance according to the authors' expert estimate.

For example, at Štúrovo railway station there were altogether 38 vehicles weighed on the wagon weighbridge in 2017 [19,20], however, ŽSR was asked for shunting by few carriers in that year. This is because shunting services are not included in the first access package of ŽSR. This claim would increase the costs of the infrastructure manager for shunting crews which would also be manifested in higher costs of shunting for the carrier. Last but not least, individual companies hire their shunters and wagon supervisors from each other at this station, depending on orders. Therefore number 9 is chosen in the pair-wise comparison of criteria. This way other decisions of authors could be described, too.

**Table 7.** Pair-wise comparison of variants according to individual criteria.

| Variant | Weight | Variant |
|---|---|---|
| | *Comparison of variants according to criteria 1* | |
| A | 9 8 7 6 5 4 3 2 1 2 3 4 5 6 7 8 9 | B |
| A | 9 8 7 6 5 4 3 2 1 2 3 4 5 6 7 8 9 | C |
| B | 9 8 7 6 5 4 3 2 1 2 3 4 5 6 7 8 9 | C |
| | *Comparison of variants according to criteria 2* | |
| A | 9 8 7 6 5 4 3 2 1 2 3 4 5 6 7 8 9 | B |
| A | 9 8 7 6 5 4 3 2 1 2 3 4 5 6 7 8 9 | C |
| B | 9 8 7 6 5 4 3 2 1 2 3 4 5 6 7 8 9 | C |
| | *Comparison of variants according to criteria 3* | |
| A | 9 8 7 6 5 4 3 2 1 2 3 4 5 6 7 8 9 | B |
| A | 9 8 7 6 5 4 3 2 1 2 3 4 5 6 7 8 9 | C |
| B | 9 8 7 6 5 4 3 2 1 2 3 4 5 6 7 8 9 | C |
| | *Comparison of variants according to criteria 4* | |
| A | 9 8 7 6 5 4 3 2 1 2 3 4 5 6 7 8 9 | B |
| A | 9 8 7 6 5 4 3 2 1 2 3 4 5 6 7 8 9 | C |
| B | 9 8 7 6 5 4 3 2 1 2 3 4 5 6 7 8 9 | C |

In the next stage of the first step, the same comparison of criteria is realized for carriers which utilize these services. It will be a comparison by the importance and frequency of utilization of the service by each carrier. That is the content of Table 7.

The second step contains the formation of the Saaty's decision matrix. On the main diagonal the values will equal one, because individual alternatives are compared with themselves here. The other four values (it will be a 4 × 4 matrix) above the main diagonal are determined by the given entity in the comparison. The comparison and assignment of weights is usually determined as follows: the alternative which is located in a column is compared to an element in a top row. The values below the main diagonal will be written as reciprocals of individual weights above their main diagonal according to Equation (2) [21,22].

$$Values\ below\ the\ main\ diagonal = \frac{1}{values\ above\ the\ main\ diagonal} \tag{2}$$

In the first stage of the second step, the Saaty's matrix will be formed for the provided services (Table 8).

**Table 8.** Saaty's decision criterion matrix.

| Criterion | A | B | C | D |
|---|---|---|---|---|
| A | 1 | 7 | 9 | 2 |
| B | 1/7 | 1 | 9 | 1/6 |
| C | 1/9 | 1/9 | 1 | 9 |
| D | 1/2 | 6 | 1/9 | 1 |

In the second stage of the second step, the Saaty's matrix will be created for railway undertakings which utilize services at Štúrovo railway station. A demonstration example is presented in Table 9.

**Table 9.** Saaty's decision matrix of variants.

| Variant | A | B | C |
|---|---|---|---|
| | *Comparison of variants according to criteria 1* | | |
| A | 1 | 1/2 | 7 |
| B | 2 | 1 | 8 |
| C | 1/7 | 1/8 | 1 |

**Table 9.** *Cont.*

| Variant | A | B | C |
|---|---|---|---|
| *Comparison of variants according to criteria 2* | | | |
| A | 1 | 6 | 9 |
| B | 1/6 | 1 | 8 |
| C | 1/9 | 1/8 | 1 |
| *Comparison of variants according to criteria 3* | | | |
| A | 1 | 1 | 1 |
| B | 1 | 1 | 1 |
| C | 1 | 1 | 1 |
| *Comparison of variants according to criteria 4* | | | |
| A | 1 | 1/3 | 8 |
| A | 3 | 1 | 8 |
| B | 1/8 | 1/8 | 1 |

The third step is characterized by the determination of an eigenvector of the matrix ($X_K$) and a normalized eigenvector of the matrix ($X_{KN}$) according to Equation (3).

$$v_i = \sqrt[n]{a_{i1} \times a_{i2} \times a_{i3} \times a_{i4}} \qquad (3)$$

where:

$a_i$—a row criterion

$v_i$—an eigenvector of the matrix

$n$—a dimension of the matrix ($4 \times 4$ in our case)

In the first stage we will calculate an eigenvector of the matrix and a normalized eigenvector of the matrix for vectors of the criteria matrix in Table 10.

**Table 10.** Variant matrix vectors.

| Line Number | Custom Matrix Vector | Normed Eigenvector of a Matrix |
|---|---|---|
| $v_1$ | 3.35 | 0.62 |
| $v_2$ | 0.68 | 0.13 |
| $v_3$ | 0.58 | 0.11 |
| $v_4$ | 0.76 | 0.14 |
| Σ | **5.37** | **1** |

The second stage of the third step lies in creating the vectors of the matrix by individual criteria which are listed in Table 11.

**Table 11.** Vectors of the matrix of variants of individual criteria.

| Line Number | Custom Matrix Vector | Normed Eigenvector of a Matrix |
|---|---|---|
| *Comparison of variants according to criteria 1* | | |
| $v_1$ | 1.52 | 0.45 |
| $v_2$ | 1.44 | 0.43 |
| $v_3$ | 0.38 | 0.12 |
| Σ | **3.34** | **1** |
| *Comparison of variants according to criteria 2* | | |
| $v_1$ | 4.33 | 0.61 |
| $v_2$ | 2.52 | 0.35 |
| $v_3$ | 0.26 | 0.04 |
| Σ | **7.11** | **1** |

**Table 11.** *Cont.*

| Line Number | Custom Matrix Vector | Normed Eigenvector of a Matrix |
| --- | --- | --- |
| *Comparison of variants according to criteria 3* | | |
| $v_1$ | 1 | 0.34 |
| $v_2$ | 1 | 0.33 |
| $v_3$ | 1 | 0.33 |
| $\Sigma$ | **3** | **1** |
| *Comparison of variants according to criteria 4* | | |
| $v_1$ | 1.39 | 0.30 |
| $v_2$ | 2.88 | 0.64 |
| $v_3$ | 0.25 | 0.06 |
| $\Sigma$ | **4.52** | **1** |

Then the fourth step follows, which is focused on the calculation of an eigenvalue of the matrix and the biggest eigenvalue of the matrix. The calculation of the eigenvalue of the matrix is done using Equation (4).

$$\lambda_i = \frac{a_{i1} \times w_1 + a_{i2} \times w_2 + a_{i3} \times w_3 + a_{i4} \times w_4}{w_i} \tag{4}$$

where:

$\lambda_i$—custom matrix number,
$a_i$—i-th row of matrix A,
$w$—normed eigenvector of a matrix A.

Equation (5) serves for the calculation of the biggest value of the eigenmatrix. For the sake of clarity, the values of the eigenvalue of the matrix as well as the biggest value of the eigenmatrix will be presented within one table.

$$\lambda_{max} = \frac{1}{n} \times (\lambda_1 + \lambda_2 + \lambda_3 + \lambda_4) \tag{5}$$

where:

$\lambda_{max}$—the biggest value of the matrix,
$n$—a dimension of the matrix ($4 \times 4$ in our case),
$\lambda_i$—an eigenvalue of the matrix (in a respective row).

The first stage of the fourth step will comprise Table 12, where eigenvalues of the matrix and the biggest value of the eigenmatrix will be processed to evaluate the services provided at railway stations.

**Table 12.** Matrix values.

| Eigenvalue of the Matrix | | The Largest Eigenvalue of the Matrix |
| --- | --- | --- |
| $\lambda_1$ | 2.93 | |
| $\lambda_2$ | 1.37 | 4.61 |
| $\lambda_3$ | 12.04 | |
| $\lambda_4$ | 2.08 | |

The second stage of the fourth step comprises the calculation of the eigenvalue of the matrix and the biggest value of the eigenmatrix for individual railway undertakings utilizing the railway station services. Table 13 presents the individual values.

**Table 13.** Matrix values according to individual criteria.

| Eigenvalue of the Matrix | | The Largest Eigenvalue of the Matrix |
|---|---|---|
| *Comparison of variants according to criteria 1* | | |
| $\lambda_1$ | 2.43 | |
| $\lambda_2$ | 2.25 | 1.97 |
| $\lambda_3$ | 1.22 | |
| *Comparison of variants according to criteria 2* | | |
| $\lambda_1$ | 4.34 | |
| $\lambda_2$ | 2.40 | 2.63 |
| $\lambda_3$ | 1.13 | |
| *Comparison of variants according to criteria 3* | | |
| $\lambda_1$ | | |
| $\lambda_2$ | 1.67 | 1.67 |
| $\lambda_3$ | | |
| *Comparison of variants according to criteria 4* | | |
| $\lambda_1$ | 1.96 | |
| $\lambda_2$ | 2.25 | 1.78 |
| $\lambda_3$ | 1.12 | |

In the final fifth stage, there will be an AHP decision matrix formed. This matrix is the final decision matrix and, based on the results presented in it, the order of importance of individual services for selected carriers will be set. Afterwards there will be some measures proposed which should be enacted for the modernization of Štúrovo railway station in order to ensure a non-discriminatory access to services.

The decision matrix contains the following indicators:

- criteria—four services provided by Štúrovo railway station,
- weights of criteria—values of the normalized eigenvector of the matrix from Table 10,
- importance for the entities of railway transport—values of the normalized eigenvector of the matrix from Table 11,
- the weighted sum—it is calculated as a sum of the product of weights and the measure of importance of individual railway undertakings,
- the order—the order of utilizing individual services by railway undertakings will be determined by the number of won points.

Table 14 shows the final evaluation of the railway station's result using the AHP method. The use of this method should give the most accurate results.

**Table 14.** Saaty's decision criterion matrix.

| Criterion | Weight | Level of Importance for Railway Undertaking | | |
|---|---|---|---|---|
| | | **A** | **B** | **C** |
| **A** | 0.62 | 0.45 | 0.43 | 0.11 |
| **B** | 0.13 | 0.61 | 0.35 | 0.04 |
| **C** | 0.11 | 0.33 | 0.33 | 0.33 |
| **D** | 0.14 | 0.31 | 0.31 | 0.06 |
| **Weight sum** | | **0.438** | **0.438** | **0.118** |
| **Position** | | **1.** | **2.** | **3.** |

It is clear from the table that the carriers ZSSK CARGO and PSŽ have an important position at Štúrovo railway station. On the other hand, the position of Metrans Danubia, a.s., is of a much smaller weight. This is also caused by the fact that for this carrier the priority border crossing is in Komárno–Komárom, and the border crossing Štúrovo–Szob is used as a diversion only in case of closures or other emergency situations.

The biggest value is attributed to the service of stabling sidings which in spite of its "popularity" is not sufficient and shift dispatchers of carriers are forced to utilize other stations, or tell half-truths to persuade train dispatchers of the infrastructure manager to side-track their train in particular. In contrast, shunting services provided by the infrastructure manager feature the smallest weight. These services are utilized by few carriers due to their financial unprofitability.

## 4. Conclusions

The AHP method is one of the most accurate decision methods of the multi-criteria analysis [23]. As much as possible it removes subjectivity, and despite a comparatively higher difficulty of calculation it produces relevant data. A considerable factor, bringing an element of simplification into the calculation process, is the application of software tools [24,25]. The methodology of non-discriminatory capacity allocation using the AHP method was applied to the Štúrovo railway station. The reason is that this frontier exchange station does not meet capacity requirements. The methodology mentioned in the article can be applied to any station in the ŽSR network. It is worth mentioning that a cluster analysis was used on the BDŽ network in passenger transport [26,27]. It would be interesting to use this method in further research.

The method also brought some achievements related to the issue of the utilization of railway transport services at Štúrovo railway station. The most important thing is the position of individual carriers which reflects the order of their importance. This fact is demonstrated in Figure 2.

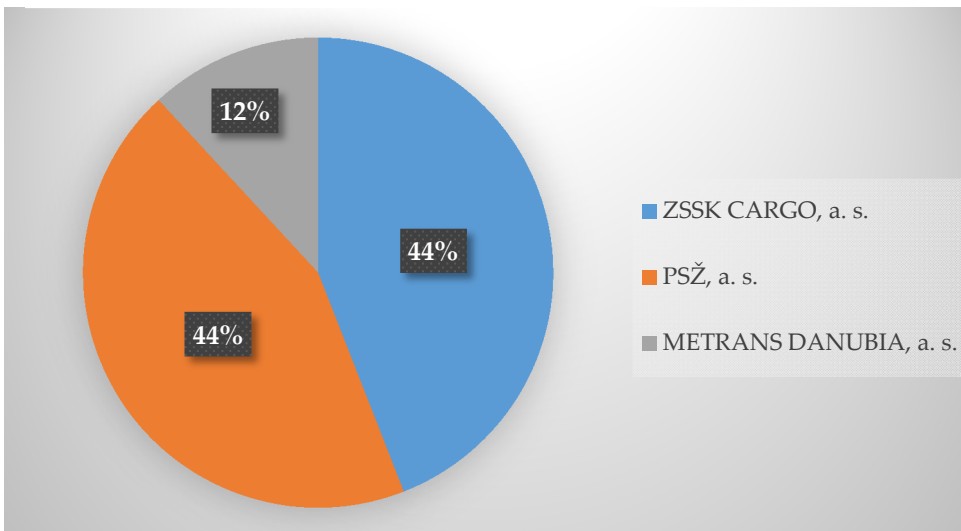

**Figure 2.** Position of carriers.

The equal first position is shared by ZSSK CARGO and PSŽ with a total value of 0.44, which makes 44% when translated to their performances. These numbers are not supported with any official statistical data, they only represent experts' estimates. On the other hand, the position of the Metrans Danubia carrier is much weaker; it achieved 0.12, i.e., 12% performance. The reason is that this carrier does not utilize this border crossing with priority.

An equally significant part of the investigation is the weight which is assigned to services provided by Štúrovo railway station. The status of service utilization by carriers mentioned above is perfectly demonstrated in Figure 3.

Per the chart, the most utilized service is the service of stabling sidings which is an everyday reality at this railway station. In contrast, the shunting service offered by the infrastructure manager is utilized to the smallest extent. The main reason is that transport companies either have their own employees designated for the activity or they are able to provide this service among themselves.

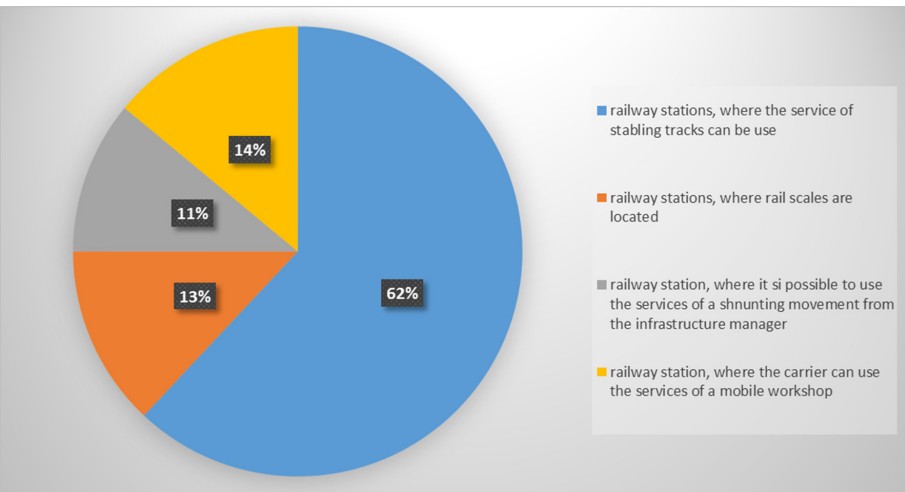

**Figure 3.** Rate of use of services.

This implies that in the case of modernization of Štúrovo railway station, the priority should lie in increasing the number of station tracks which could be utilized by carriers.

**Author Contributions:** A.S. conceived conceptualization and designed the methodology; performed the resources, calculations and investigations of the data; E.B.—writing review and editing; E.B.—supervision, project administration and funding acquisition. All authors have read and agreed to the published version of the manuscript.

**Funding:** This paper is a partial output of the grant project VEGA 1/0509/19: Optimizing the use of railway infrastructure with support of modal split forecasting.

**Conflicts of Interest:** The authors declare no conflict of interest.

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
