# Peer review of "Problems of Access to Services at Railway Stations in Freight Transport in the Slovak Republic"

_sustainability, doi:10.3390/su12198018_

Round 1
Reviewer 1 Report
Although the topic is timely and interesting, the paper in its current form needs some major revision in the introduction part, to examine the literature on previous studies.
- P1-Line,17 I suggest authors to highlight the importance of the Multi critieria decisn making and then connect it with the present study.
- P1-Line,29, In second paragarph, here I suggest authros to highlight what previous studies have used AHP and other MDCN and why AHP method is appropriate for this study.For example. https://doi.org/10.1016/j.promfg.2018.10.082
- AHP method is well know and many studies have been conducted.-I would expect from the authors to go beyond just giving the definitions of the constructs of proposed models, but also providing information on whether these models or may be separate parts of the models have been used in previous studies.
- --Moreover, the literature cited is not exhaustive and is relatively not recent. You might want to increase the number of works cited, explain in detail what they find and what are their actual limitations. Moreover, you should add more recent studies on the topic.
- -In heading literature review, I encourage authors to cite more literature which has been used AHP or different modelling approaches.
- https://doi.org/10.1016/j.promfg.2019.02.088
- DOI: https://doi.org/10.1007/978-3-319-71062-4. 978-3-319-71062-4
- https://doi.org/10.1016/j.promfg.2018.10.082
- Results analysis is very long, and it seems ambiguous. It can not get reader attention. I suggest authors not to put more focus on writing literature, instead, support your results with the previously established literature and give your arguments.
Author Response
Response to Reviewer 1 Comments
Point 1: Suggest authors to highlight the importance of the Multi critieria decisn making and then connect it with the present study.
Response 1: Thank you for your comment. Multicriteria analysis has been used in rail transport at various levels. In passenger transport and with the infrastructure manager. In (1) the issue is being discussed in the Balkan region in twelve different rail markets. The methodology is based on multi-criteria assessment of the level of railway development. The approach presented in this paper could help railway companies to make decisions about railway transport services. The results show that the criteria: maximum train technical speed (13%), ERTMS Level (10%), number of train kilometres per year (9%) and Ro-La intermodal service (9%) have a great importance in the ranking. It was found that the most developed railway transports in the Balkan region are Turkey, Croatia, Slovenia, and Romania. In our article, based on the AHP method, three criteria are also selected, the importance of which is assessed at a specific railway station from the perspective of carriers.
- A MULTI-CRITERIA ASSESSMENT APPROACH FOR THE EVALUATION OF RAILWAY TRANSPORT IN THE BALKAN REGION. Stoilova, Svetla. 6, 2019, PROMET-TRAFFIC & TRANSPORTATION, s. 655-668. ISSN 0353-5320.
Point 2: Results analysis is very long, and it seems ambiguous. It can not get reader attention. I suggest authors not to put more focus on writing literature, instead, support your results with the previously established literature and give your arguments.
Thank you for your comments. The references was added, thank you for your interest reference.

Reviewer 2 Report
In the reviewed paper Authors presented the issue of access of freight carriers to services in railway stations in the Slovac Republic. As the rail freight market is liberalized, the number of carriers is gradually increasing. In this situation, infrastructure capacity is often insufficient. Therefore, it necessary to set the order of access to services in railway stations. This can be done using various methods of multicriteria analysis. In the reviewed paper Authors used the Analytic Hierarchy Process method in order to solve this problem. Paper is interesting, but in some places, unfortunately, needs some refinement. In my opinion, paper can be published, after taking into account the following remarks:
- in "Abstract" section, Authors did not briefly present the results obtained in the paper. Abstracts should give a pertinent overview of the paper and should summarize the paper's main findings. This should be improved,
- "Reference" section needs to be formatted as required by the Sustainability journal,
- "Author Contributions" needs to be formatted as required by the Sustainability journal,
- In "Introduction" section, Authors didn't place the study topic in a broad context. In Introduction section, the current state of the research field should be reviewed carefully and key publications cited. In this case, a short paragraph should be added to the Introduction section in which the problem of freight transport in the broad context of freight transport in various modes of transport will be presented. Authors should write in this short paragraph, how important is the role of freight transport/goods transport in various modes of transport nowadays, f.ex. this kind of transport must be carried out on time, under certain conditions, often in special containers, etc., f.ex .: (1) Gao, Y., Zou, X., Chen, R., Ma, Y., Li, C., Zhang, Y.: Freight Mode Coordination in China: From the Perspective of Regional Differences. Sustainability 2020, 12(7), 2996; (2) Macioszek E.: Freight Transport Planners as Information Elements in the Last Mile Logistics. [in:] Integration as Solution for Advanced Smart Urban transport Systems. Advances in Intelligent Systems and Computing 844. Springer International Publishing Switzerland 2019, p. 242-251; (3) He, Z., Haasis, H.D.: A Theoretical Research Framework of Future Sustainable Urban Freight Transport for Smart Cities. Sustainability 2020, 12(5), 1975; (4) Zou, X., Somenahall, S., Scrafton D.: Evaluation and analysis of urban competitiveness and spatial evolution. International Journal of Logistics Research and Applications, 2019; (5) Macioszek E.: First and last mile delivery - problems and issues. [in:] Advanced Solutions of Transport Systems for Growing Mobility. Advances in Intelligent Systems and Computing 631. Springer International Publishing Switzerland 2018, p. 147-154; (6) He, Z.: The challenges in sustainability of urban freight network design and distribution innovations: a systematic literature review. International Journal of Physical Distribution & Logistics Management, 2020,
- at the end of the Introduction section, Authors should briefly describe what is contained in each section of the paper,
- in Table 1, the acronym "j. s. c." should be explained and replaced in full name in English if applicable,
- in Table 6 and Table 7, Authors used a red colour in order to indicated some values. In the paper text, or under this tables in legend, Authors should explain the meaning of this red colour,
- the paper gives the impression of a paper written in a hurry, hence it is underdeveloped in some places, e.g. Table 2 is divided into 2 pages. The same remark is dedicated for Table 3, it should be improved,
- "Figure 1. Decision making process" has no relationships between individual elements (arrows), which means no dependencies exist between these elements. It should be improved,
- Authors should explain whether the obtained results are confirmed in real conditions?
- There is no discussion section in the paper text,
- The Conclusion section is written in a very general way. Authors should conduct a detailed discussion of the obtained results in the Conclusion section (or add a discussion section).
Author Response
Response to Reviewer 2 Comments
Point 1: In "Abstract" section, Authors did not briefly present the results obtained in the paper. Abstracts should give a pertinent overview of the paper and should summarize the paper's main findings. This should be improved.
Response 1: The article deals with the issue of access of freight carriers to services in railway stations. With the liberalization of the rail freight market, the number of carriers is gradually increasing. In this situation, infrastructure capacity is often insufficient. Therefore, it is necessary to set the order of access to services in railway stations. The article will use the process of analytical hierarchy as one of many methods of multicriteria analysis. Three important indicators will be selected for carriers: railway stations, where the service of stabling tracks can be used, railway stations, where rail scales are located, railway stations, where it is possible to use the services of a shunting movement from the infrastructure manager, railway stations, where the carrier can be using the services of a mobile workshop. At the end of the article, the order of access to these services will be compiled according to the order of importance for railway undertakings. A significant factor will also be an approximate quantification of the performance of individual carriers passing through the selected station.
Point 2: In Table 6 and Table 7, Authors used a red colour in order to indicated some values. In the paper text, or under this tables in legend, Authors should explain the meaning of this red colour.
Response 2: In Table 6, red is used for pairwise comparison of the criteria listed in Table 5. In Table 7, red is used to express the importance for pairwise comparison of criteria by variant. In both cases, the red color emphasizes the degree of importance according to the authors' expert estimate.
Point 3: The Conclusion section is written in a very general way. Authors should conduct a detailed discussion of the obtained results in the Conclusion section (or add a discussion section).
Response 3: The methodology of non-discriminatory capacity allocation using the AHP method was applied to the Štúrovo railway station. The reason is that this frontier exchange station does not meet capacity requirements. The methodology mentioned in the article can be applied to any station on the ŽSR network.
Thank you very much for your suggestions.

Round 2
Reviewer 1 Report
I appreciate the author handling the reviewer comments and now manuscript has improved a lot from the previous version, still some minor issues needed to take into consideration.
In the introduction, you need to connect the state of the art to your paper goals. Please follow the literature review by a clear and concise state of the art analysis. This should clearly show the knowledge gaps identified and link them to your paper goals. Please reason both the novelty and the relevance of your paper goals. In the conclusions, in addition to summarising the actions taken and results, please strengthen the explanation of their significance.
-Also, provide a table by highlighting the previous studies on AHP in the similar domain and their findings in the literature review section.
Author Response
Thank you for your comment. No one has dealt with the issue of evaluation of railway stations using the AHP method in freight transport. AHP method was used to evaluate railway infrastructure (citation 16). Cluster analysis was used to evaluate railway stations in terms of passenger transport (citation 26).

Reviewer 2 Report
Authors improved paper according to reviewer comment.
Paper can be published in the present form.
Author Response
Thank you for your comment. We are pleased that you are interested in the article. Your comments helped improve our article. Thanks for the help.
